# Subcutaneously Injectable Hyaluronic Acid Hydrogel for Sustained Release of Donepezil with Reduced Initial Burst Release: Effect of Hybridization of Microstructured Lipid Carriers and Albumin

**DOI:** 10.3390/pharmaceutics13060864

**Published:** 2021-06-11

**Authors:** Nae-Won Kang, So-Yeon Yoon, Sungho Kim, Na-Young Yu, Ju-Hwan Park, Jae-Young Lee, Hyun-Jong Cho, Dae-Duk Kim

**Affiliations:** 1College of Pharmacy and Research Institute of Pharmaceutical Sciences, Seoul National University, Seoul 08826, Korea; nwkangkr@snu.ac.kr (N.-W.K.); sokyu123@naver.com (S.-Y.Y.); sh122333@snu.ac.kr (S.K.); nyyu8520@snu.ac.kr (N.-Y.Y.); cyclone5pjh@naver.com (J.-H.P.); 2Dong-A Pharm. Research Laboratories, Youngin 17073, Korea; 3College of Pharmacy, Chungnam National University, Daejeon 34134, Korea; jaeyoung@cnu.ac.kr; 4College of Pharmacy, Kangwon National University, Chuncheon 24341, Korea; hjcho@kangwon.ac.kr

**Keywords:** hyaluronic acid, hydrogel, microstructured lipid carrier, human serum albumin, sustained release, reduced initial burst release, Alzheimer’s disease, donepezil

## Abstract

The daily oral administration of acetylcholinesterase (AChE) inhibitors for Alzheimer’s disease features low patient compliance and can lead to low efficacy or high toxicity owing to irregular intake. Herein, we developed a subcutaneously injectable hyaluronic acid hydrogel (MLC/HSA hydrogel) hybridized with microstructured lipid carriers (MLCs) and human serum albumin (HSA) for the sustained release of donepezil (DNP) with reduced initial burst release. The lipid carrier was designed to have a microsized mean diameter (32.6 ± 12.8 µm) to be well-localized in the hydrogel. The hybridization of MLCs and HSA enhanced the structural integrity of the HA hydrogel, as demonstrated by the measurements of storage modulus (G′), loss modulus (G″), and viscosity. In the pharmacokinetic study, subcutaneous administration of MLC/HSA hydrogel in rats prolonged the release of DNP for up to seven days and reduced the initial plasma concentration, where the C_max_ value was 0.3-fold lower than that of the control hydrogel without a significant change in the AUC_last_ value. Histological analyses of the hydrogels supported their biocompatibility for subcutaneous injection. These results suggest that a new hybrid MLC/HSA hydrogel could be promising as a subcutaneously injectable controlled drug delivery system for the treatment of Alzheimer’s disease.

## 1. Introduction

Alzheimer’s disease is an irreversible and progressive neurodegenerative disease that slowly destroys memory, language, executive function, and personality [1]. The use of acetylcholinesterase (AChE) inhibitors is the first treatment option [2], and they are administered daily in the form of oral dosages or as a transdermal patch. Studies on the other routes, including nasal delivery, were also reported [3,4]. However, it is sometimes difficult for patients to regularly take them without the help of others; this can lead to low efficacy or overdose-related toxicities owing to their narrow therapeutic index [5]. Thus, it is necessary to develop a formulation for sustained release of AChE inhibitors as an alternative to once-daily administration.

The hydrogel is a three-dimensional biopolymer network with tunable physical and mechanical properties [6,7,8,9]. It has been used for sustained drug delivery by hybridization with other drug carriers such as polymeric microspheres [10] and lipid carriers [11]. The long polymer structure acts as the framework of the formulation and retains the particles inside the gel network, thereby physically protecting them from elimination and/or degradation. These kinds of hydrogel/particle hybrids achieved sustained drug release by which the dosing interval can be increased [12,13].

Among several drug carriers, solid lipid-based particles have been commonly used for the preparation of sustained drug release systems due to their excellent biocompatibility and desirable physicochemical properties for topical skin, oral, and parenteral administration [14]. As the drugs are homogenously dispersed in a solid matrix, they can be also protected from contact with the environment. Thus, solid lipid-based carriers have been hybridized with hydrogels for the development of long-acting formulations [11,15]. This also improves their viscoelastic properties, providing higher structural integrity [16]. The particle size of a lipid carrier is an important factor to be considered when designing hybrid hydrogel formulations. Particles with a microsized diameter are more advantageous for localization inside the hydrogel network [17]. Lipid-based carriers of nanosized particles can easily leak out of the hydrogel, resulting in low drug loading capacity, high initial burst release of the drug, and particle aggregation [18].

Although the sustained drug release can be achieved through lipid carrier-hybridized hydrogels, the increase of the toxicity risk by the initial burst release still remains. Lipid-based carriers have commonly exhibited the initial burst release of drugs due to high drug concentration near the particle surface, large surface area of the particles, and drug adsorption on the surface [19,20,21,22,23]. This problem can cause severe toxicities for the drugs with a narrow therapeutic index, including AChE inhibitors. Thus, the strategies not only for prolonging the release, but also for reducing the initial burst release are necessary to improve the safety and efficacy of drugs in the formulation of hydrogels.

The objective of this study was to develop a hyaluronic acid (HA) hydrogel hybridized with microstructured lipid carriers (MLCs) and human serum albumin (HSA) for sustained release of the AChE inhibitor donepezil (DNP) as a model drug after subcutaneous injection. HA was selected as a biopolymer of the hydrogel owing to its excellent biocompatibility and rheological suitability for subcutaneous injection [24]. The composition and preparation of the MLCs were optimized, considering that hybridized MLCs would afford both a sustained-release effect and high localization inside the hydrogel structure. Moreover, HSA was additionally hybridized in the HA hydrogel structure to reduce the initial burst release of DNP and partially contribute to sustained release. HSA is expected to retain the release of DNP from the hydrogel by reversible protein binding, thereby serving as a drug reservoir (Figure 1).

## 2. Materials and Methods

### 2.1. Materials

Sodium phosphate monobasic dihydrate, triethylamine, phosphoric acid, human serum albumin, oleic acid, olive oil, peanut oil, and carbamazepine were supplied by Sigma-Aldrich Co. (St. Louis, MO, USA). Miglyol 812 was obtained from Oleochemicals (Putrajaya, Malaysia). Mineral oil was provided by SAMCHUN Chemicals (Seoul, Korea). Bio sodium hyaluronate (150–600 kDa) was purchased from SK Bioland Co., Ltd. (Cheonan, Korea). Donepezil, stearic acid, lecithin, and formic acid were provided by TCI Co. (Tokyo, Japan). Methanol (MeOH) and acetonitrile (ACN) were acquired from Fisher Scientific Korea Ltd. (Seoul, Korea). Phosphate-buffered saline (PBS) was purchased from Mediatech (Manassas, VA, USA).

### 2.2. Solubility of DNP in a Liquid Lipid

The solubility of DNP in various oils (i.e., oleic acid, Miglyol 812, olive oil, peanut oil, soy lecithin, mineral oil) was determined to select a liquid lipid for the preparation of MLCs. Briefly, 10 mg DNP was added to 1 mL of each oil and vortex-mixed for 30 min, followed by sonication for 30 min. Then, the samples were incubated in a water bath at 37 °C for 72 h under gentle agitation (50 rpm). After centrifugation at 13,200 rpm for 5 min, the supernatant was diluted with ACN, and the concentration of DNP was analyzed using a high-performance liquid chromatography (HPLC) system equipped with a reversed-phase C-18 column (150 × 4.6 mm, 5 μm; Fortis Technologies, Ltd., Cheshire, UK) and a guard column (Fortis C18; 100 Å; 5 μm; 10 × 4.0 mm). The mobile phase comprised 0.2 M phosphate buffer:MeOH:triethylamine (50:50:0.5 (*v/v/v*)). The flow rate was 1.0 mL/min and the injection volume was 20 µL. The UV detection wavelength was set at 268 nm. The retention time of DNP was 7 min and the total run time was 10 min. The lower limit of quantification (LLOQ) of the calibration curve was 200 ng/mL.

### 2.3. Optimization and Characterization of Microstructured Lipid Carriers (MLCs)

Stearic acid (melting point: about 70 °C) was selected as a solid lipid phase for the preparation of MLCs, while oleic acid was selected as a liquid lipid based on the solubility study (Appendix A). The MLCs were fabricated with different weight ratios of oleic acid to stearic acid to optimize the composition. Briefly, DNP (20 mg), stearic acid (90 mg), and oleic acid (0~20 mg) were completely dissolved in 1 mL acetone in a 70 °C water bath. Then, the solution was dropped into 3 mL deionized distilled water (DDW) with stirring (500 rpm) at room temperature and left for 10 min until the lipid particles were solidified. After centrifugation at 16,000× *g* for 5 min, the supernatant was aspirated to obtain the lipid particle pellet, which was then reconciled in 2 mL DDW with vortex-mixing. This washing process was repeated twice to remove the organic solvent and residual components. Then, tip sonication was applied for 90 s to homogeneously disperse MLCs (two seconds on, three seconds off; probe-type sonicator; Vibra-Cell VC 750 ultrasonic processor; Sonics & Materials, Newtown, CT, USA).

The particle size distribution of MLCs was analyzed using a laser diffraction particle size analyzer (Microtrac, S3500, Microtrac Inc., Montgomeryville, PA, USA). Zeta potential of MLCs dispersed in 0.1× PBS was determined using the electrophoretic light scattering method (ELS-Z, Otsuka Electronics, Tokyo, Japan). To determine the content of DNP in MLCs, an aliquot (5 mg) of MLC powder was weighed after lyophilization of aqueous dispersion. Then, DNP was extracted with 5 mL of ACN by vortex-mixing for 1 h, followed by sonication for 1 h. After centrifugation at 13,200 rpm for 5 min, the supernatant was diluted 100 times with ACN. The concentration of DNP was analyzed by using HPLC, as described above. The DNP content (%, *w/w*) and preparation recovery (%, *w/w*) of each MLC were calculated using the following equations:(1)$$\mathrm{Content}\;(\%,\;w/w)=\frac{Amount\;of\;DNP\;in\;MLCs}{Total\;amount\;of\;MLCs}\;\times \;100$$
(2)$$\mathrm{Recovery}\;(\%,\;w/w)=\frac{Actual\;amount\;of\;DNP\;in\;MLCs}{Theoretical\;amount\;of\;DNP\;in\;MLCs}\;\times \;100$$

Powder X-ray diffraction (pXRD) and differential scanning calorimetry (DSC) studies were conducted to investigate the crystal state of DNP in MLCs. For comparison, a physical mixture of stearic acid and donepezil (4:6, *w/w*) was prepared by vortex-mixing of each component, while blank MLCs were fabricated without DNP. The pXRD pattern of MLCs was investigated using D8 Advance with DAVINCI (BRUKER AXS Inc., Karlsruhe, Germany) equipped with a Cu Kα_1_ radiation source (λ = 1.5418 Å). The acceleration voltage and tube current were 40 kV and 40 mA, respectively. Each sample was analyzed at a scanning speed of 1 s/step. DSC study was performed using DSC-Q1000 (TA Instrument, New Castle, DE, USA). Each sample was heated from 25 to 200 °C at a heating rate of 10 °C/min under a nitrogen atmosphere (50 mL/min) using aluminum sealed pans. Molecular interaction in MLCs was assessed using a Fourier-transform infrared (FT-IR) spectrophotometer (Nicolet 6700; Thermo Scientific, Waltham, MA, USA) with the attenuated total reflection (ATR) mode. All the spectra were scanned at room temperature with 32 scans in the range from 650 to 4000 cm^−1^ with a resolution of 4 cm^−1^.

### 2.4. Preparation of the MLC/HSA Hydrogel

The HA hydrogel hybridized with MLCs and HSA (MLC/HSA hydrogel) was prepared in two steps: (1) MLC preparation and (2) gelation with HA. Briefly, MLCs were prepared as described above to contain DNP (20 mg), stearic acid (90 mg), and oleic acid (10 mg) (Table 1). Then, HSA (100 mg) was dissolved in the aqueous suspension (2 mL) of MLCs. Finally, HA (120 mg) was added and completely blended for 30 min, followed by storing overnight at room temperature to remove bubbles. The “MLC hydrogel” was prepared by the same method above, except the addition of HSA. The “blank hydrogel” was prepared by simply dissolving HA in DDW without DNP, while the “control hydrogel” was added with DNP into the “blank hydrogel” at the same amount as above.

### 2.5. Optical and Microscopic Morphologies of the MLC/HSA Hydrogel

To investigate the morphological change of the hydrogels over time, 1 mL of each hydrogel was passed through a syringe (18 G) and placed on a plastic plate at room temperature. The change of visual appearance over time was observed by taking pictures immediately after ejection and after 1 h. The morphology of hydrogels was also observed by field-emission scanning electron microscopy (FE-SEM; JSM-7800F Prime, JEOL Ltd., Tokyo, Japan). The hydrogels were lyophilized and then coated for 60 s with platinum by ion sputtering for the FE-SEM analysis.

### 2.6. Rheological Analysis

Viscoelasticity of the hydrogels was visually compared by inverting the vials containing 4 mL of each hydrogel. Optical appearance of the hydrogels in vials was obtained at 0, 1, and 6 h to assess the degree of flow. Rheological properties of the hydrogels were also investigated using a rheometer (Advanced Rheometric Expansion System, Rheometric Scientific, Inc., Piscataway, NJ, USA). The equipment measured the storage modulus (G′) and loss modulus (G″) profiles of the hydrogels with two measuring modes at 25 °C: angular frequency (w)-dependent (1~100 rad/s) at the fixed strain of 1% and strain (γ)-dependent (1~100%) at the fixed frequency of 2 rad/s. Frequency-dependent dissipation factor (tan δ, calculated from G″/G′) and viscosity (Pa•s) were measured at the fixed strain of 1%.

### 2.7. In Vitro Release Study of DNP and HSA

The in vitro release profiles of DNP from the hydrogels were observed using the dialysis cellulose membrane sac (Spectra/Por^®^; 12–14 kDa molecular weight cutoff; Spectrum Medical Devices, Fort Mill, SC, USA). Aliquots (100 µL) of the hydrogels were loaded into the cellulose membrane sac, after which it was immersed in the release medium (PBS, 30 mL) at 37 °C and agitated at 50 rpm. Aliquots (0.5 mL) of the release medium were collected at predetermined times (5 min, 15 min, 30 min, 1 h, 2 h, 4 h, 6 h, 9 h, 24 h, 48 h, 72 h, 96 h, 120 h, 144 h, and 168 h), and the same volume of the fresh medium was replenished at each time point. In the DNP solution group, 100 µL DNP in PBS (5 mg/mL) was added in the dialysis membrane sac as the control. The amount of DNP released was analyzed using the HPLC method as described above.

The in vitro release of HSA was studied the same as DNP, except that a dialysis cellulose membrane with a larger molecular cutoff (300 kDa) was used due to large molecular weight of HSA (66.5 kDa). In the HSA solution group, 100 µL HSA in PBS (50 mg/mL) was added in the dialysis membrane sac as the control. Concentration of BSA was analyzed using a BCA assay kit (Pierce; Rockford, IL, USA) according to the manufacturer’s protocol.

### 2.8. Pharmacokinetics

The pharmacokinetics of DNP after a subcutaneous injection of hydrogel was investigated in Sprague Dawley (SD) rats. Male rats of 250~350 g were obtained from ORIENT BIO Inc. (Seongnam, Korea). The rats were reared in a light-controlled room at 22 ± 2 °C and relative humidity of 55 ± 5% in the Animal Center for Pharmaceutical Research, College of Pharmacy, Seoul National University (Seoul, Korea). The study protocol was approved by the Institutional Animal Care and Use Committee of the Seoul National University (Seoul, Korea). After the rats were anesthetized with an intramuscular injection of Zoletil (Virbac, Carros, France) (50 mg/kg), each hydrogel was injected subcutaneously (10 mg/kg as DNP) to the back of rats. In the DNP solution group, 10 mg/kg DNP in PBS (25 mg/mL) was subcutaneously injected as the control. Blood samples (~300 µL) were collected from the tail vein at the predetermined time points (15 min, 30 min, 1 h, 2 h, 4 h, 6 h, 9 h, 24 h, 48 h, 72 h, 96 h, 120 h, 144 h, and 168 h). The collected blood samples were centrifuged at 16,000× *g* for 2 min at 4 °C, and an aliquot (50 µL) of the supernatant was collected. They were vortex-mixed with ACN (150 µL) containing carbamazepine (as the internal standard; 200 ng/mL) for 5 min, and the mixtures were centrifuged at 16,000× *g* for 5 min. The supernatant (120 µL) of the samples was transferred into a sample vial.

The plasma concentration of DNP in the samples was analyzed using an HPLC system (1260 Infinity; Agilent Technologies, Palo Alto, CA, USA) equipped with an Agilent Technologies 6430 Triple Quad LC–MS/MS system (Agilent Technologies, Wilmington, DE, USA), a G1367E autosampler, a G1312B binary pump, a G1316C thermostatted column compartment, and a G1330B thermostat. An aliquot (2 µL) of the sample was injected into a Poroshell 120 EC-C18 column (50 mm × 4.6 mm, 2.7 μm; Agilent Technologies, Wilmington, DE, USA) with a guard column (Agilent EC-C18; 5 × 4.6 mm; 2.7 μm). Isocratic elution was carried out at a flow rate of 0.4 mL/min. The mobile phase was comprised of 0.2% (*v/v*) formic acid in DDW:ACN (3:7, *v/v*). The mass spectrometric detection was performed in the positive electrospray ionization (ESI) mode. The optimized ESI source parameters were manually set as follows: gas temperature, 300 °C; gas flow, 11 L/min; nebulizer pressure (nitrogen), 15 psi; and capillary voltage, 4.00 kV. The multiple reaction monitoring (MRM) mode was used at the unit resolution for both Q1 and Q3 mass filters. The optimized molecular mass transitions of the precursor to product ion/fragmentor voltages (V)/collision energies (eV) for DNP and carbamazepine were m/z 380.3 → 91.3/165/44 and m/z 237.0 → 194.0/126/18, respectively. The data were acquired and processed using MassHunter Workstation Software (version B.05.00; Agilent Technologies, Wilmington, DE, USA). Pharmacokinetic parameters, including area under the plasma drug concentration/time curve (AUC), maximum plasma concentration (C_max_), time to reach C_max_ (T_max_), terminal half-life (T_1/2_), and mean residence time (MRT), were calculated using the WinNolin^®^ software (build 8.1.0.3530; Certara, Princeton, NJ, USA).

### 2.9. Histology

After the subcutaneous injection of hydrogels in rats, they were humanely sacrificed by cervical dislocation at the predetermined time points (6 h, 1 day, 2 day, 4 day, and 7 day), and then the hydrogels were separated by dissecting the skin. The optical images of hydrogels were taken to observe the change of size and appearance. For the histological study, the skin region where the hydrogel was injected was dissected on day 7, after which these samples were immediately fixed in a neutral buffered formalin solution for 24 h. After dehydrating with ethanol, the skin was embedded in paraffin. Sections with the thickness of 4 μm were stained with hematoxylin and eosin (H&E) or Masson’s trichrome (MT) according to the standard protocol and were then observed under a microscope. The collagen area after MT staining was quantified using a Teledyne Lumenera Infinity3-1 camera and the Infinity Analyze program (Ottawa, ON, Canada).

### 2.10. Statistical Analysis

Statistical analyses of the data were performed using GraphPad Prism 8.0.2 (GraphPad Software Inc., San Diego, CA, USA). The data were analyzed using one-way analysis of variance (ANOVA), followed by Tukey’s multiple comparisons test. The data were presented as the means ± standard deviation (SD), and *p*-value < 0.05 was considered significantly different.

## 3. Results

### 3.1. Optimization and Characterization of MLCs

Appendix A shows the solubility of DNP in various liquid lipids. As the highest solubility (183.5 mg/mL) was observed in oleic acid, it was selected for the preparation of MLCs. Stearic acid was selected as the solid lipid based on its melting point (approximately 70 °C) and biocompatibility [25,26]. MLCs were fabricated with various weight ratios of oleic acid/stearic acid, and their physicochemical properties were assessed to optimize the composition (Table 2). MLCs with the particle size of approximately 30 µm were obtained when the weight ratio of oleic acid increased up to 10% (*w/w*). However, the particle size increased markedly when 15% and 20% oleic acid was added to the particles. As reported in a previous study [27], an excessive amount of oleic acid could disturb the formation of a dense lipid matrix, leading to a loose structure and low stability of the particles. It has been reported that oleic acid has a strong negative charge under physiological conditions (i.e., in PBS) due to carboxylic acid in the polar head [28]. Thus, the addition of oleic acid resulted in a highly negative zeta potential of MLCs, suggesting that oleic acid was well-embedded in the solid lipid matrix (i.e., the stearic acid matrix). As the content and recovery of DNP were also the highest (14.6 ± 0.4% and 87.6 ± 2.4%, respectively) at 10% (*w/w*) oleic acid, it was selected as the optimum composition of MLCs and was further characterized.

The particle size of the optimized MLCs was 32.6 ± 12.8 µm with monomodal distribution (Figure 2A). The crystallographic properties of DNP in the optimized MLCs were characterized by analyzing the pXRD and DSC patterns. As shown in Figure 2B, the crystalline state of DNP exhibited a distinctive peak pattern that was not observed in MLCs, whereas MLCs exhibited a pattern similar to that of stearic acid and blank MLCs (i.e., MLCs without DNP). These results indicate that the crystalline state of DNP was transformed to an amorphous state and/or dissolved in the lipid matrix; this is further supported by the endothermic peaks in the DSC analysis (Figure 2C). The crystalline state of DNP and stearic acid exhibited their own endothermic peaks at 92.6 °C and 70.1 °C, respectively. However, MLCs and blank MLCs exhibited a peak only at 64.1 °C, which was attributed to stearic acid, and the endothermic peak of DNP disappeared. The lower shift of the stearic acid peak in blank MLCs could have resulted from the embedment of oleic acid into the solid lattice of stearic acid. It has been reported that oleic acid could disturb the intact crystallization of stearic acid, leading to a lower endothermic peak [29]. It may be noted that the physical mixture of stearic acid and DNP exhibited the glass transition temperature (Tg) of DNP together with the peak of stearic acid. During the DSC measurement, the stearic acid powder turned into the liquid state at a temperature of 70.1 °C (melting point of stearic acid), whereas the DNP powder maintained the solid state up to 92.6 °C. The liquid state of stearic acid could partially dissolve the DNP powder and affect its crystalline state, which in turn shows the glass transition temperature of DNP. In contrast, DNP in MLCs was clearly dissolved and/or dispersed in liquid stearic acid and oleic acid during the preparation process at 70 °C, thereby resulting in the disappearance of the endothermic peak of DNP. Thus, the results of the pXRD and DSC studies strongly confirm that DNP in MLCs was dissolved and/or well-dispersed as an amorphous state inside the particles.

Molecular interactions between DNP and lipid components in MLCs were investigated using FT-IR analysis (Figure 2D). The functional groups in the DNP’s chemical structure, such as ether, aromatic ring, piperidine, and carbonyl group, exhibited distinctive absorption bands at 1120 cm^−1^ (C–O–C stretching), 1310 cm^−1^ (C–H wagging), 1440 cm^−1^ and 1500 cm^−1^ (C–N–C stretching), 1590 cm^−1^ (C–C=C stretching in the aromatic ring), and 1680 cm^−1^ (C=O stretching). Of note, MLCs exhibited absorption bands similar to those of blank MLCs. Disappearance of absorption bands of DNP in MLCs indicates that DNP was dissolved/dispersed in MLCs and not adsorbed to the particle surface.

### 3.2. Preparation and Morphology of MLC/HSA Hydrogels

Table 1 summarizes compositions of the hydrogels prepared in this study. After fabricating MLCs in DDW, HSA and HA were added to the suspension to form the MLC/HSA hydrogel. The optical appearance of the hydrogels immediately after passing through a disposable syringe is shown in Figure 3A. While the blank and control hydrogels exhibited a transparent and homogenous gel structure, the MLC-containing hydrogels (i.e., the MLC hydrogel and the MLC/HSA hydrogel) were yellowish. They were more rigid than the hydrogels without MLCs (i.e., the blank hydrogel and the control hydrogel) and temporarily retained the thread-like structure immediately after ejection. However, they changed to the homogenous hydrogel structure within 1 h under ambient conditions (Figure 3B), which indicated that they would be restored by autonomous interactions after the subcutaneous injection.

Microscopic images of the MLC/HSA hydrogel were observed using SEM (Figure 3C–F). The blank hydrogel exhibited a fiber-like structure with void spaces (Figure 3C), while the porosity of the hydrogel decreased markedly with the addition of HSA (Figure 3D). Hybridization of HSA seemed to build a denser reticular structure owing to the electrostatic interactions between HSA and HA. In the MLC hydrogel, the structure of MLCs was clearly observed inside the porous HA structure (Figure 3E). The inner structure of the MLC/HSA hydrogel exhibited homogeneous dispersion of MLCs in the denser structure of the hydrogel network (Figure 3F), which would help retain MLCs inside the hydrogel.

### 3.3. Rheological Analysis

Rheological properties of the hydrogels were investigated by observing their optical appearance and evaluating their rheological parameters (Figure 4). All the prepared hydrogels (as in Table 1) featured viscoelasticity, holding their volume at the bottom of the vials when they were inverted (Figure 4A). However, blank and control hydrogels flowed down within 1 h, which indicated that the addition of DNP did not affect viscoelasticity of the HA hydrogel. Meanwhile, the MLC hydrogel and the MLC/HSA hydrogel remained for up to 1 h and 6 h, respectively. These results imply that the addition of MLCs and HSA into the HA hydrogel rendered the structure more rigid and significantly decreased its flowability.

Viscoelastic behavior of the hydrogels was assessed quantitatively by measuring the storage modulus (G′) and the loss modulus (G″) as the function of angular frequency (Figure 4B) and strain sweep (Figure 4C). In all the hydrogels, the value of G′ was higher than that of G″, which indicated they behave more like an elastic solid rather than a liquid. The frequency-dependent dissipation factor (tan δ, G″/G′) was also consistent with these results, featuring a value lower than one over the angular frequency range of 1–100 rad/s (Figure 4C). Moreover, the values of both G′ and G″ of the hydrogels were in the following order: control hydrogel < MLC hydrogel < MLC/HSA hydrogel. These results indicate that elasticity of the hydrogel increased significantly with the addition of MLCs (i.e., the MLC hydrogel) and that it further increased with the addition of HSA (i.e., the MLC/HSA hydrogel). Of note, the MLC/HSA hydrogel exhibited the lowest tan δ value (Figure 4D) and the highest viscosity value over the entire frequency range (Figure 4E), which confirmed the enhanced elastic properties. The increased viscosity indicates a higher resistance of the hydrogels against deformation in response to stress [30]. Thus, the rheological analyses showed that the hybridization of MLCs and HSA enhanced the viscoelastic properties of the HA hydrogel, which would provide structural integrity after injection into the skin.

### 3.4. In Vitro Release Study of DNP and HSA

Figure 5A shows the in vitro release profiles of DNP from the hydrogels. As the DNP solution group exhibited rapid and complete drug release (100% released within 2 h), it was confirmed that the dialysis membrane did not hinder the release of DNP into the medium. The in vitro release of DNP from the control hydrogel was relatively rapid compared to the other hydrogels; the cumulative amounts released in 9 h and 24 h were 68.2% and 98.2%, respectively. However, the MLC hydrogel exhibited a slower release profile compared to the control hydrogel; the cumulative amounts released in 9 h and 24 h were 45.4% and 66.3%, respectively. Thus, the incorporation of DNP into MLCs slowed down the release of DNP from the hydrogel. The large particle size of MLCs (32.6 µm, Table 2) also seems to have contributed to the retention of particles in the hydrogel network, thereby releasing DNP for a longer period of time. It may be noted that the release rate of DNP from the MLC/HSA hydrogel was further reduced compared to the other hydrogels; the cumulative amounts released in 9 h and 24 h were 34.6% and 49.0%, respectively. Notably, the initial DNP released from the MLC/HSA hydrogel was reduced significantly compared to that of the MLC hydrogel. Moreover, the reduction in the initial burst release was proportional to the HSA content in the MLC/HSA hydrogel (Appendix A). As the protein binding of DNP was reported to be high (~75%) [31], the intermolecular interaction between free DNP (i.e., the DNP released from MLCs) and HSA could temporarily retain DNP before the release into the medium, thereby reducing the initial burst release rate. It may also be noted that the in vitro release of DNP from the MLC/HSA hydrogel reached a plateau after two days (i.e., less than 100% release) for up to seven days (Figure 5A). As the molecular cutoff of the dialysis was not large enough for HSA to be released into the medium, it is expected that HSA would accumulate in the dialysis membrane sac during the release study. Although this condition was useful for evaluating the effect of HSA on reducing the initial burst release of DNP, some portion of DNP intermolecularly bound to high concentrations of HSA could not be released into the medium. This assumption was supported by comparing the amount of DNP released in 48 h from the MLC/HSA hydrogels with different amounts of HSA (Appendix A). When 50, 75, or 100 mg/mL of HSA were added to the MLC hydrogel, the amount of DNP released in 48 h decreased proportionally from 61.3 ± 8.30% to 47.5 ± 0.5%. Moreover, when the in vitro release of HSA from the MLC/HSA hydrogel was studied using a dialysis cellulose membrane with a larger molecular cutoff (300 kDa), HSA was slowly released from the MLC/HSA hydrogel for up to seven days (Figure 5B). The rapid and complete release of HSA from the HSA solution group confirmed that the dialysis membrane was not a barrier to the release of HSA. Thus, it is expected that HSA would be completely released from the MLC/HSA hydrogel in vivo, which in turn completes the release of DNP. Moreover, the slow release of HSA from the MLC/HSA hydrogel also partially contributed to the sustained release of DNP from the hydrogel (Figure 1).

### 3.5. Pharmacokinetics

Figure 6 shows the time-dependent plasma concentration profiles of DNP after the subcutaneous injection of various hydrogels (10 mg/kg as DNP) in rats, and the pharmacokinetic parameters are summarized in Table 3. The pharmacokinetic parameters of the control hydrogel were not significantly different from those of the DNP solution. Although the control hydrogel exhibited a slightly delayed T_max_ value, the elimination rate of DNP (T_1/2_ ≈ 4 h) was similar to that of the DNP solution group, and the plasma concentration was below the detection limit after two days. This result is consistent with previous studies showing that the mesh size of typical biomedical hydrogels is not suitable for retarding the release of the drug incorporated in the gel network [17]. On the other hand, the MLC-hybridized hydrogels exhibited sustained release profiles, which is consistent with the in vitro release study (Figure 5A). The half-life values (T_1/2_) of the MLC hydrogel (14.6 h) and the MLC/HSA hydrogel (28.8 h) were significantly higher than that of the control hydrogel (4.2 h). It may be noted that HSA hybridization extended the time to reach the maximum plasma concentration (T_max_) value, whereas the time for the MLC/HSA hydrogel (24 h) increased significantly, by more than three times compared to the MLC hydrogel (6.8 h). Moreover, the MLC/HSA hydrogel exhibited a significantly lower C_max_ value (204 ng/mL) and a longer MRT value (34.3 h) than the MLC hydrogel (348 ng/mL and 22.3 h, respectively). As there was no significant difference in AUC values of the hydrogels and the DNP solution, the relative bioavailability (relative BA) was close to 100%, indicating that DNP was released completely from the hydrogels. Thus, the hybridization of HSA into the MLC hydrogel provided sustained release and reduced the initial burst release of DNP.

### 3.6. Histology

Optical images of hydrogels separated from the tissue at a predetermined time after the subcutaneous injection are shown in Figure 7A. As the control hydrogel had low viscosity and rigidity, the structure rapidly degraded within two days. However, the structures of the MLC hydrogel and the MLC/HSA hydrogel were well-preserved for up to four days and seven days, respectively, which confirmed the structural integrity and self-healing property of the hybrid hydrogels in subcutaneous tissues. Moreover, the subcutaneous tissues near the MLC/HSA hydrogel did not exhibit any sign of bleeding 6 h, 2 d, and 7 d after the injection, which additionally supports its biocompatibility (Appendix A).

Figure 7B shows the histology of the skin seven days after the hydrogel injection to investigate any damage in the injected region. In the H&E staining study, no notable inflammatory cells were observed in the dermal or subcutaneous layers of the skin after injection of each hydrogel. MT staining suggested no significant changes in the collagen area, including voids or breaks, between the hydrogel groups (Appendix A). These results imply that the hydrogels prepared in this study are biocompatible for subcutaneous injection without causing tissue damage, including inflammation and collagen content.

## 4. Discussion

HA has been widely investigated in drug delivery systems owing to its excellent biocompatibility and rheological properties for subcutaneous injection [24]. However, one of the major limitations of the HA hydrogel as a drug delivery system is the rapid release of drugs from the gel network. Thus, diverse approaches have been reported to control the drug release rate from the HA hydrogel, which includes chemical modification, physical and covalent crosslinking. Among them, hybridization of solid lipid particles has been suggested as an alternative strategy which does not require chemical crosslinkers and conjugation [11]. In this study, we prepared the HA hydrogel hybridized with MLCs and HSA and investigated the effect on the sustained release and reduced the initial burst release of DNP.

The composition and particle size of MLCs are the most essential factors to be optimized for the controlled release of drugs. Stearic acid, a non-toxic and biocompatible fatty acid, was chosen as a solid lipid due to its suitable physicochemical properties for drug delivery [25]. As its melting point is approximately 70 °C, it maintains the solid structure when administered in the body, resulting in a sustained drug release pattern by slow diffusion and dissolution [32,33]. The composition of MLCs was optimized with oleic acid to achieve the highest content and recovery of DNP (Table 2). The optimized MLCs exhibited the particle size of 32.6 ± 12.8 µm with monomodal distribution (Figure 2A). Given the typical mesh size of a hydrogel is known to be 5–100 nm, MLCs would provide enough steric hindrance to immobilize particles in the HA network [34]. The negative zeta potential (−34.2 ± 1.5 mV), which is mainly due to the deprotonation of oleic acid, would also contribute to the formation of repulsive forces between particles and prevent their aggregation, thereby improving the chemical stability of the particles [35].

While the blank and control hydrogels had a transparent and homogeneous gel structure, the addition of MLCs and HSA resulted in more rigid and yellowish hydrogels with self-healing properties (Figure 3). The self-healing behavior is defined as the ability of materials to repair the properties to their original state after being damaged, which could enhance structural integrity and prolong the lifespan of hydrogels [36]. As no external stimuli (e.g., heat and self-healing agents) were applied to the hydrogels, it was expected that autonomous interactions led to self-healing phenomena in the hydrogels. The intermolecular interactions between MLCs and HSA markedly improved viscoelastic properties. As the lipid components in MLCs had a polar head, the intermolecular interactions between HA and MLCs could be formed by electrostatic interactions and/or hydrogen bonding. Additional hybridization with HSA (i.e., the MLC/HSA hydrogel) further increased the elasticity and viscosity of the hydrogel (Figure 4). It has been reported that interactions between HA and solid lipid-based carriers formed physical crosslinks and increased viscoelasticity [16]. In addition, as the electrostatic interaction between HA and HSA have also been reported [36,37], the hybridization of MLCs and HSA enhanced the structural integrity of the hydrogel to maintain it after the subcutaneous injection and prolong the lifespan of the hydrogel. These results are consistent with the morphological observation of the hydrogels by optical appearance and microscopic images using SEM (Figure 3), where the hybridization of MLCs and HSA resulted in a denser structure and homogeneous dispersion of MLCs in the hydrogel network.

HSA is a natural protein in human blood and has high aqueous solubility, non-immunogenicity, and non-toxicity [38]. It is widely accepted that serum albumin affects pharmacokinetics or pharmacodynamics of drugs owing to an albumin–drug interaction, which is called protein binding [39]. As binding is reversible, the albumin–drug complex can be used as a drug reservoir that can improve biodistribution and bioavailability of drugs [40]. Thus, we studied the effect of HSA hybridized in the HA hydrogel on the reduction in the initial burst release and sustained release of DNP. As shown in Figure 5A, the initial burst release of DNP from the MLC/HSA hydrogel decreased significantly compared to that of the MLC hydrogel, which may indicate that DNP released from MLCs was temporarily retained by HSA. Moreover, the slow and complete release of HSA from the MLC/HSA hydrogel for up to seven days (Figure 5B) would also contribute to the sustained release of DNP from the hydrogel.

The pharmacokinetic parameters of DNP after the subcutaneous injection of the hydrogels in rats consistently indicated the sustained release of DNP from the hydrogels (Figure 6 and Table 3). The MLC-hybridized hydrogels exhibited significantly extended T_1/2_, T_max_, and MRT values, without a significant difference in AUC values. Moreover, hybridized HSA further contributed to the delay in the release of DNP, leading to significantly higher T_max_ and MRT values. In addition, hybridization of HSA significantly reduced the C_max_ value, indicating the reduced initial burst release of DNP. There were several reports on the subcutaneous administration of DNP, including dual-crosslinked hydrogels and microneedle array patches [41,42,43]. However, we believe that hybridization of MLCs and HSA in an HA hydrogel is a novel approach that can achieve longer T_max_ and lower C_max_ values. Although the in vitro release of DNP from the MLC/HSA hydrogel was not completed for seven days (Figure 5A) owing to the entrapment of HSA in the dialysis sac membrane, 100% relative bioavailability confirmed that DNP would be completely released in vivo. Moreover, the MLC/HSA hydrogel was biocompatible for seven days after the subcutaneous injection (Figure 7), confirming that hybridization of MLCs and HSA into a hydrogel would be an effective approach for sustained release of drugs with reduced initial burst release.

## 5. Conclusions

A new subcutaneously injectable MLC- and HSA-hybridized hyaluronic acid hydrogel was successfully prepared for the sustained release of DNP. The composition of micrometer-sized (approximately 30 µm) MLCs was optimized, and MLCs were concluded to be well-localized in the hydrogel network. The MLC-hybridized hydrogels improved the rheological characteristics, exhibiting adequate structural integrity and fluidity. The MLC-hybridized hydrogels significantly prolonged the release of DNP in the in vitro release study, and the addition of HSA to the MLC hydrogel further decreased the release rate with reduced initial burst release. The in vivo pharmacokinetic study in rats was consistent with these results, confirmed by the increase of T_1/2_, T_max_, and MRT values with reduced C_max_ value. Thus, hybridization of MLCs and HSA into a hyaluronic acid hydrogel could be a promising strategy for a sustained drug release system with biocompatibility and could be applied to the treatment of Alzheimer’s disease.

## Figures and Tables

**Figure 1 pharmaceutics-13-00864-f001:**
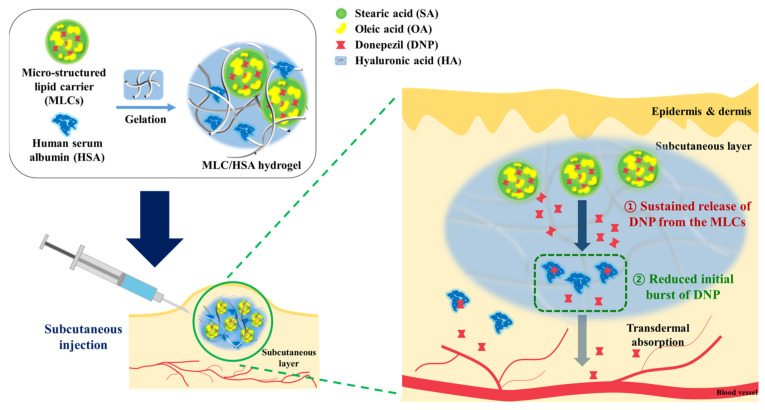
Schematic illustration of the developed hydrogel for sustained release of donepezil with the reduced initial burst release.

**Figure 2 pharmaceutics-13-00864-f002:**
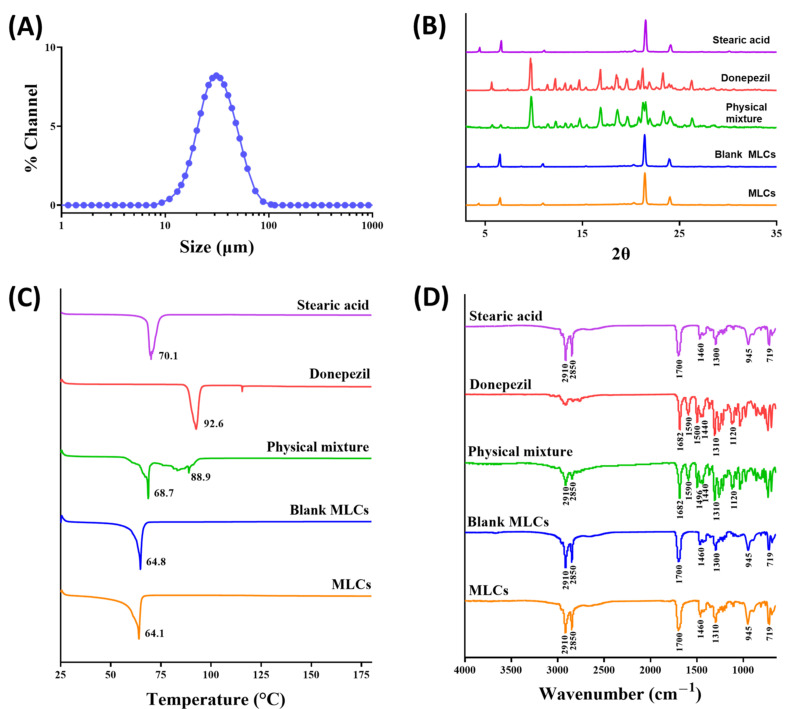
Characterization of the optimized MLCs. (**A**) Particle size distribution, (**B**) pXRD, (**C**) DSC, and (**D**) FT-IR analyses.

**Figure 3 pharmaceutics-13-00864-f003:**
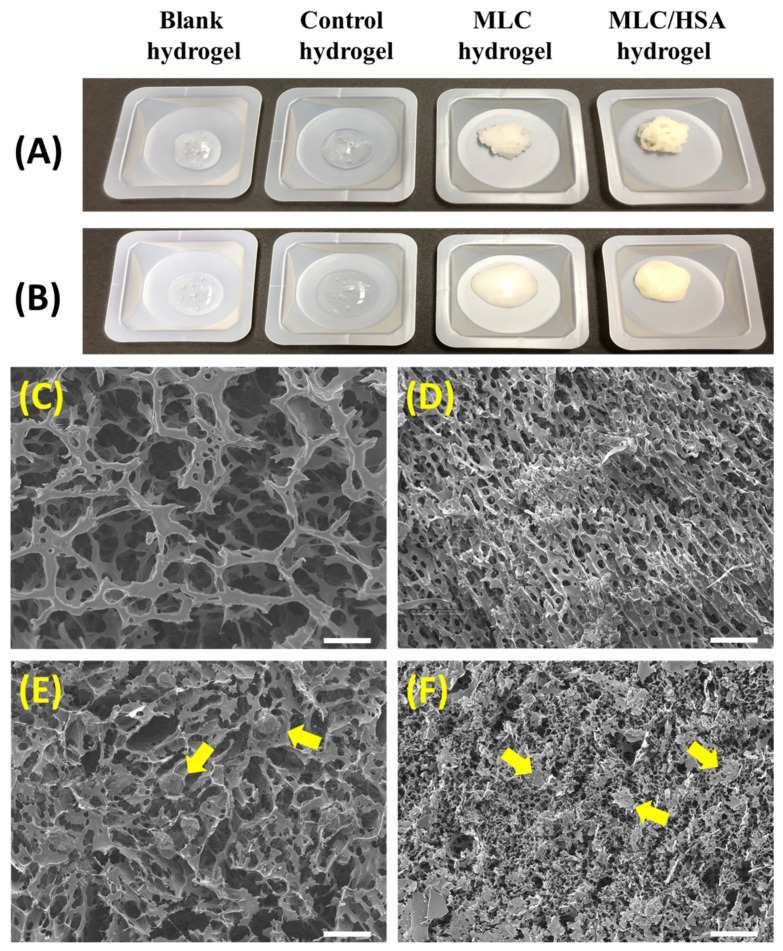
Morphology of the hydrogel formulations. (**A**) Optical appearance of the hydrogels observed immediately after passing through the syringe. (**B**) Optical appearance of the hydrogels after 1 h. FE-SEM images of (**C**) the blank hydrogel, (**D**) the blank hydrogel with albumin, (**E**) the MLC hydrogel, and (**F**) the MLC/HSA hydrogel. The length of the scale bar in SEM images is 30 µm. Yellow arrows indicate MLCs.

**Figure 4 pharmaceutics-13-00864-f004:**
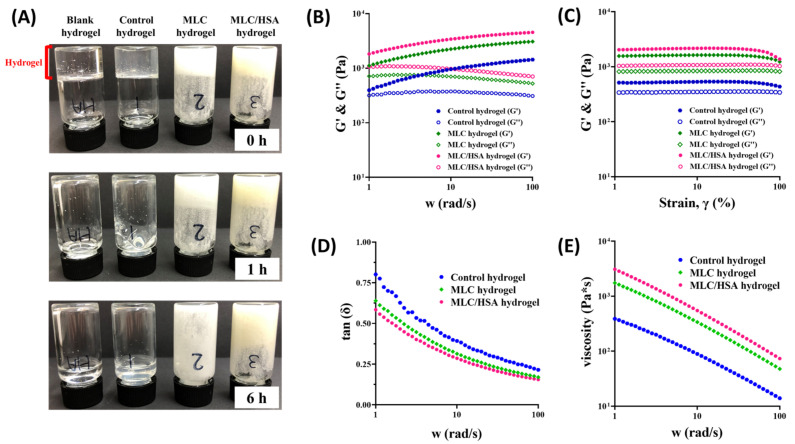
Rheological analyses of the MLC/HSA hydrogel. (**A**) Appearance of the hydrogels in the inverted vials observed immediately after inversion, as well as after 1 h and 6 h. (**B**) Storage modulus (G′) and loss modulus (G″) as the function of angular frequency (w) at 1% strain (γ), and (**C**) G′ and G″ as the function of strain sweep at the frequency of 2 rad/s. (**D**) Calculated dissipation factor (tan δ) as the function of angular frequency at 1% strain. (**E**) Viscosity (Pa•s) of the hydrogels measured by varying the angular frequency at 1% strain.

**Figure 5 pharmaceutics-13-00864-f005:**
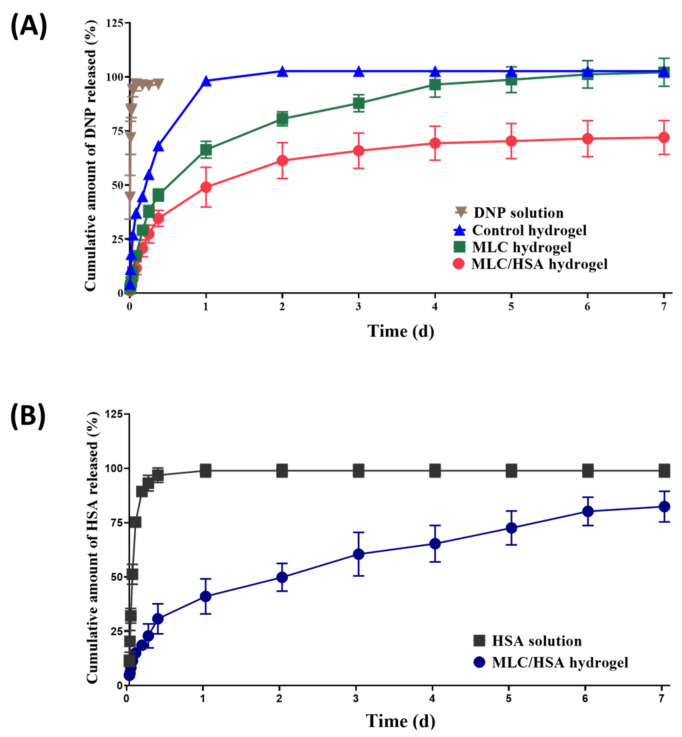
In vitro release profiles of DNP and HSA from hydrogels into PBS at 37 °C. Cumulative amount of (**A**) DNP released from the developed hydrogels and of (**B**) HSA released from the MLC/HSA hydrogel. Each point represents the mean ± SD (*n* = 4).

**Figure 6 pharmaceutics-13-00864-f006:**
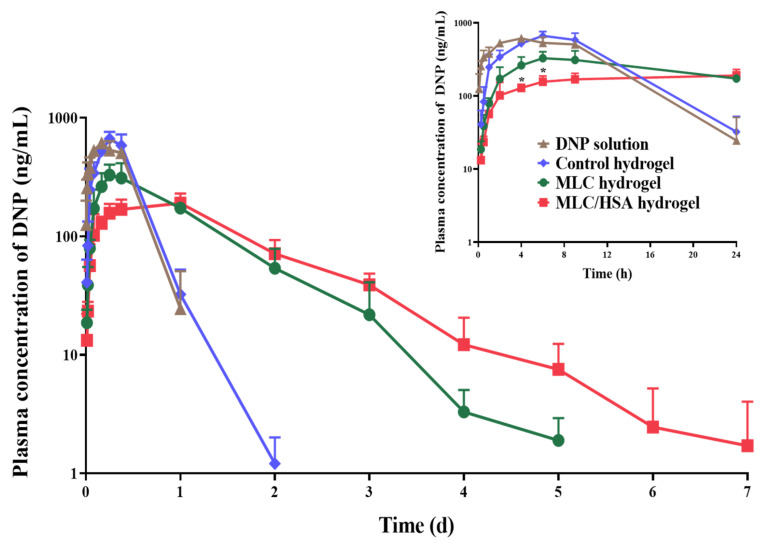
Time-dependent plasma concentration profiles of DNP after the subcutaneous injection of the developed hydrogels (10 mg/kg as DNP) in rats. The insert shows the profiles up to 24 h. Each point represents the mean ± SD (*n* = 4); * *p* < 0.05 (compared to the MLC hydrogel).

**Figure 7 pharmaceutics-13-00864-f007:**
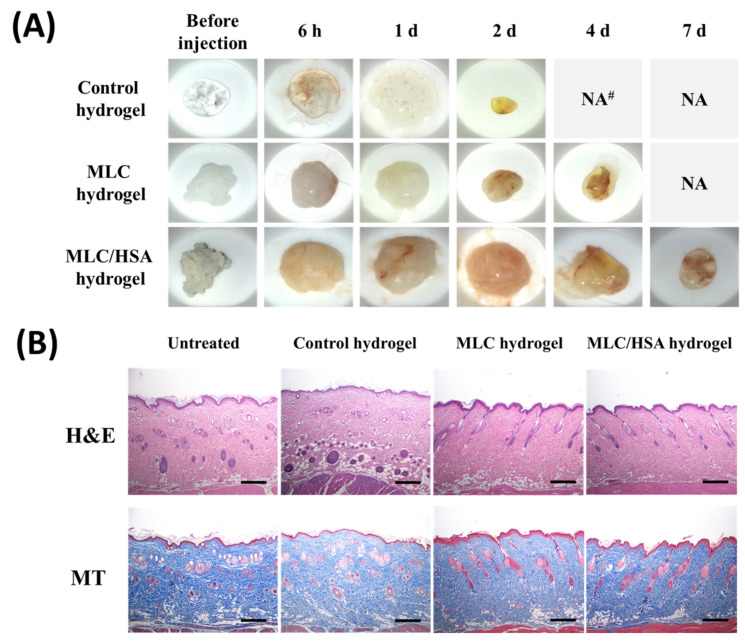
(**A**) Optical images of hydrogels after subcutaneous injection in rats. (**B**) Histology of the skin seven days after hydrogel injection. The untreated group indicates the skin without any treatment. Each group was stained with hematoxylin and eosin (H&E) or Masson’s trichrome (MT). The length of the scale bar is 400 μm.

**Table 1 pharmaceutics-13-00864-t001:** Compositions of hydrogels.

		MLCs			
Group	HA (mg)	DNP (mg)	Stearic Acid (mg)	Oleic Acid (mg)	HSA (mg)
Blank hydrogel	120	-	-	-	-
Control hydrogel	120	20	-	-	-
MLC hydrogel	120	20	90	10	-
MLC/HSA hydrogel	120	20	90	10	100

Each value is per 2 mL of DW.

**Table 2 pharmaceutics-13-00864-t002:** Physicochemical properties of MLCs with the various weight ratios of oleic acid/stearic acid.

Oleic Acid Weight ^a^ (mg)	Size Distribution (µm)	Zeta Potential (mV)	Content ^b^ (%, *w/w*)	Recovery ^c^ (%, *w/w*)
0	28.7 ± 12.2	−3.1 ± 1.2	13.2 ± 0.2	72.6 ± 1.1
5	30.0 ± 12.1	−32.5 ± 1.0	12.5 ± 0.4	71.9 ± 2.3
10	32.6 ± 12.8	−34.2 ± 1.5	14.6 ± 0.4	87.6 ± 2.4
15	180.1 ± 67.0	−45.0 ± 0.9	14.0 ± 0.3	87.5 ± 1.9
20	181.9 ± 62.9	−53.9 ± 0.3	13.2 ± 0.3	85.8 ± 2.0

^a^ Weight of oleic acid = amount (mg) of oleic acid added in the preparation of MLCs with stearic acid (90 mg) and DNP (20 mg). ^b^ Content (%, *w/w*) = $\frac{\mathrm{Amount}\;\mathrm{of}\;\mathrm{DNP}\;\mathrm{in}\;\mathrm{MLCs}}{\mathrm{Total}\;\mathrm{amount}\;\mathrm{of}\;\mathrm{MLCs}}$ × 100. ^c^ Recovery (%, *w/w*) = $\frac{\mathrm{Actual}\;\mathrm{amount}\;\mathrm{of}\;\mathrm{DNP}\;\mathrm{in}\;\mathrm{MLCs}}{\mathrm{Theoretical}\;\mathrm{amount}\;\mathrm{of}\;\mathrm{DNP}\;\mathrm{in}\;\mathrm{MLCs}}$ × 100.

**Table 3 pharmaceutics-13-00864-t003:** Pharmacokinetic parameters of DNP after the subcutaneous injection (10 mg/kg as DNP) in rats.

Parameter	DNP Solution	Control Hydrogel	MLC Hydrogel	MLC/HSA Hydrogel
T_1/2_ (h)	3.8 ± 1.1	4.2 ± 0.4	14.6 ± 2.1	28.8 ± 18.2 *
T_max_ (h)	4.5 ± 0.9	6.8 ± 1.3	6.8 ± 1.3	24.0 ± 0.0 **^,##^
C_max_ (ng/mL)	621.0 ± 42.9	671.7 ± 77.5	347.8 ± 73.6 **	203.5 ± 17.3 **^,#^
AUC_last_ (ng∙h/mL)	9003.0 ± 60.9	9356.2 ± 1336.8	9691.7 ± 1095.3	9292.5 ± 1181.9
AUC_inf_ (ng∙h/mL)	9080.7 ± 46.0	9362.3 ± 1336.9	9793.6 ± 1019.4	9410.8 ± 1021.6
MRT (h)	7.0 ± 0.4	8.4 ± 0.7	22.3 ± 4.3 **	34.3 ± 3.3 **^,##^
Relative BA^a^ (%)	100	95	107	103

^a^ Relative BA (%) = $\frac{{\mathrm{AUC}}_{\mathrm{last}}\;\mathrm{of}\;\mathrm{the}\;\mathrm{hydrogel}}{{\;\mathrm{AUC}}_{\mathrm{last}}\mathrm{of}\;\mathrm{the}\;\mathrm{DNP}\;\mathrm{solution}}$ × 100; * *p* < 0.05, ** *p* < 0.01 (compared to the control hydrogel); ^#^
*p* < 0.05, ^##^
*p* < 0.01 (compared to the MLC hydrogel). Each value is the mean ± SD (*n* = 4).

## Data Availability

The data are contained within the article or the Appendix A.

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
