# Peer review of "Subcutaneously Injectable Hyaluronic Acid Hydrogel for Sustained Release of Donepezil with Reduced Initial Burst Release: Effect of Hybridization of Microstructured Lipid Carriers and Albumin"

_pharmaceutics, 2021, doi:10.3390/pharmaceutics13060864_

Round 1

Reviewer 1 Report

General comment:

Dear authors, the manuscript titled “Subcutaneously Injectable Hyaluronic Acid Hydrogel for Sustained Release of Donepezil with Reduced Initial Burst 3 Release: Effect of Hybridization of Microstructured Lipid Carriers and Albumin” is quite interesting and the in vivo results are promising. The manuscript highlights the potential of microcarriers and hyaluronic acid hydrogels for sustained subcutaneous delivery of Donepezil.  

Comment 1: A native proofreader should revise the manuscript; some typos and grammatical mistakes must be corrected!

Comment 2: Have the authors evaluated the stability of the developed system?

Comment 3: Fig.2 presents the DSC, XRPD and FTIR spectra of the optimized MLCs, drug, lipid, physical mixture. There are actually no differences between the blank and loaded MLCs, then is that reliable to provide proof of the drug encapsulation within the MLCs?

Comment 4: In the discussion, the authors should compare the pharmacokinetics results with the previous literature on Donepezil S.C. delivery to further highlight the novelty of this work.

Comment 5: The sustained release delivery system holds high potential, especially in chronic treatment. But is it really more patient-friendly to administer the drug parentally compared to the oral route? The oral route is known to enhance the patient's adherence to the treatment, so how would the authors justify their selection of sc delivery system.

Reviewer 2 Report

This manuscript represents a good attempt to sustain the release of an Alzheimer's drug through subcutaneous injection of hydrogels. The manuscript is very well designed and written and I recommend its publication after carrying these minor changes:

1- In the introduction: Different other attempts of delivering and sustaining the drugs release through different routes for ALzheimer treatment should be briefly mentioned. The authors can refer to :

Nat Prod Res. 2018 Dec;32(24):2873-2881. doi: 10.1080/14786419.2017.1385017. 

2- In figure 2: Graph (a) is not a histogram. Please correct.

3- In figure 6: The insert is not conforming with the main graph. Please correct or comment.

Reviewer 3 Report

The manuscript by Kang et al. describes a novel study investigating the potential of a hyaluronic acid hydrogel hybridized with microstructured lipid carriers and human serum albumin for the sustained release of donepezil in the treatment of Alzheimer's disease.

I believe that both the experiment and the presentation of results are well-designed and -written.

I have a single concern regarding the novelty of the manuscript reflected in the selected references. In this context, I suggest adding more references published in the last 2 years. Some examples include:

  • 10.3390/biomedicines8100421
  • https://doi.org/10.33263/BRIAC111.84248430
  • https://doi.org/10.3390/pharmaceutics13020170
  • http://dx.doi.org/10.3390/polym12051138
